# Label-Free Liquid Chromatography–Mass Spectrometry Quantitation of Relative *N*- and *O*-Glycan Concentrations in Human Milk in Japan

**DOI:** 10.3390/ijms25031772

**Published:** 2024-02-01

**Authors:** Toshiyuki Yamaguchi, Hirofumi Fukudome, Junichi Higuchi, Tomoki Takahashi, Yuta Tsujimori, Hiroshi M. Ueno, Yasuhiro Toba, Fumihiko Sakai

**Affiliations:** 1Milk Science Research Institute, Megmilk Snow Brand Co., Ltd., 1-1-2 Minamidai, Kawagoe-shi 350-1165, Saitama, Japan; toshiyuki-yamaguchi@meg-snow.com (T.Y.); h-fuku@meg-snow.com (H.F.); junichi-higuchi@meg-snow.com (J.H.); hiroshi-ueno@meg-snow.com (H.M.U.); 2Department of Research and Development, Bean Stalk Snow Co., Ltd., 1-1-2 Minamidai, Kawagoe-shi 350-1165, Saitama, Japan; tomoki-takahashi@beanstalksnow.co.jp (T.T.); yuta-tsujimori@beanstalksnow.co.jp (Y.T.); y-toba@beanstalksnow.co.jp (Y.T.)

**Keywords:** human milk, *N*-glycan, *O*-glycan, human milk oligosaccharide, secretor status, liquid chromatography–mass spectrometry

## Abstract

Human milk is abundant in carbohydrates and includes human milk oligosaccharides (HMOs) and *N*/*O*-glycans conjugated to proteins. HMO compositions and concentrations vary in individuals according to the maternal secretor status based on the fucosyltransferase 2 genotype; however, the profile of *N*/*O*-glycans remains uninvestigated because of the analytical complexity. Herein, we applied a label-free chromatography–mass spectrometry (LC–MS) technique to elucidate the variation in the composition and concentration of *N*/*O*-glycans in human milk. We used label-free LC–MS to relatively quantify 16 *N*-glycans and 12 *O*-glycans in 200 samples of Japanese human milk (1–2 months postpartum) and applied high performance anion exchange chromatography with pulsed amperometric detection to absolutely quantify the concentrations of 11 representative HMOs. Cluster analysis of the quantitative data revealed that *O*-glycans and several HMOs were classified according to the presence or absence of fucose linked to galactose while *N*-glycans were classified into a different group from *O*-glycans and HMOs. *O*-glycans and HMOs with fucose linked to galactose were more abundant in human milk from secretor mothers than from nonsecretor mothers. Thus, secretor status influenced the composition and concentration of HMOs and *O*-glycans but not those of *N*-glycans in human milk.

## 1. Introduction

Human milk provides optimal nutrition for infants because of the bioactive substances present in addition to basic nutrients such as protein, fat, minerals, and vitamins. Human milk is abundant in carbohydrates, and these contain approximately 6% lactose and 0.5–2% human milk oligosaccharides (HMOs) in mature milk, which exceeds the concentration of total milk protein. Infants can digest lactose into glucose and galactose that are then absorbed in the small intestine as a vital infant energy source but cannot digest HMOs. HMOs instead have many biological functions, such as preventing pathogens from invading the intestinal epithelium [1,2,3], stimulating the growth of beneficial intestinal bacteria (e.g., *Bifidobacterium*, *Bacteroides*, and *Lactobacillus*) [4,5], developing the infant brain [6], and regulating infant immune systems [7]. HMOs are complex glycans of which >150 have so far been identified and structurally characterized; however, not every woman synthesizes the same set of HMOs. The HMO composition depends on the expression of α1,2-fucosyltransferase (FUT2), which is encoded by the *Se* gene [8]. Individuals with an active *Se* locus are classified as secretors. The milk of secretor women is abundant in 2′-fucosyllactose (2′-FL), lacto-*N*-fucopentaose-I (LNFP I), and other α1,2-fucosylated HMOs, although that of nonsecretor women that lack a functional FUT2 enzyme does not contain α1,2-fucosylated HMO [9]. Additionally, other factors such as the lactation period, ethnic group, and nutritional and environmental aspects can affect the concentration and composition of HMOs [10,11,12,13].

Other than HMOs, carbohydrates in human milk exist in glycoconjugates, principally glycoproteins. The majority of human milk proteins (up to 70%) are heavily glycosylated [14] and include lactoferrin, α-lactalbumin, secretory immunoglobulin A, κ-casein, and several proteins of the milk fat globule membrane. The glycan moieties of these glycoproteins contribute to protein folding, biological recognition, and protection of proteins from digestion and have also been recently shown to protect infants against pathogens [15] and serve as substrates for bifidobacterial growth in the infant gut [16]. Glycoprotein glycans are covalently attached to proteins primarily through *N*- and *O*-linked glycosylation, with the former occurring on asparagine residues and the latter occurring on serine or threonine residues. *N*-glycans are classified as high-mannose, complex, and hybrid types depending on their monosaccharide composition and branching. The *O*-glycans comprise eight *O*-glycan core types, and the nonreducing ends of *N*/*O*-glycans are frequently fucosylated and sialylated. Thus, *N*/*O*-glycans are highly structurally diverse. The structural diversity of *N*/*O*-glycans is presumed to vary among individuals, similar to the variation seen among HMOs, but the details are unknown. Human milk has been analyzed for the *N*-glycans [15,17,18,19,20,21,22,23], *O*-glycans [24], or both *N*/*O*-glycans [25,26,27] present. Although human milk was provided by 4–30 women for these reports, no information concerning individual differences or the distribution of *N*/*O*-glycans was available because the individual milk samples were pooled before analysis. Therefore, to investigate individual differences in *N*/*O*-glycans, it is necessary to quantify *N*/*O*-glycans in each milk sample from a larger number of individuals. Furthermore, no studies concerning *N*/*O*-glycans in the milk of Japanese women have yet been conducted.

Liquid chromatography (LC)–mass spectrometry (MS) and LC–fluorescence detection (FLD) are valuable techniques in the identification and quantitation of *N*/*O*-glycans. Analysis of *N*/*O*-glycans using LC–MS or LC–FLD requires two pretreatment steps: the release of *N*/*O*-glycan from glycoproteins and the derivatization of released *N*/*O*-glycans using a fluorescent reagent [28]. Performing these pretreatments on each of a large number of human milk samples is complex and time-consuming and consequently limits the sample size. Therefore, a simple and rapid pretreatment method is required. Previous reports showed that peptide-*N*-glycosidase F (PNGase F) could release *N*-glycans from glycoprotein for quantitative analysis via LC–MS without derivatization [17,18]. For *O*-glycans, Kameyama et al. developed eliminative oximation that enables β-elimination of *O*-glycans from glycoproteins using hydroxylamine and 1,8-diazabicyclo[5.4.0]undec-7-ene (DBU) to suppress *O*-glycan degradation [29]. The *O*-glycans released as a result of this procedure were analyzed using LC–FLD after labeling with 2-aminobenzoate and must also be analyzed using LC–MS without labeling because the *N*-acetylgalactosamine (GalNAc) at the reducing end of the *O*-glycan immediately formed an oxime upon release from the glycoprotein; the subsequent molecules are anticipated to be more ionized by electrospray ionization than native oligosaccharides. If the released *N*/*O*-glycans can be quantitatively analyzed via LC–MS without derivatization, the *N*/*O*-glycans in multiple specimens can be simply and rapidly analyzed.

Therefore, the purpose of this study was to elucidate the variation in the composition and concentration of *N*/*O*-glycans associated with those of HMOs in human milk using a label-free LC–MS technique. We relatively quantified *N*/*O*-glycans using label-free LC–MS methods and absolutely quantified the concentrations of representative HMOs via high performance anion exchange chromatography with pulsed amperometric detection (HPAE-PAD) in human milk samples from 200 healthy women in a cohort study in Japan (The Japanese Human Milk Study) [30]. We performed a cluster analysis using quantitative data of *N*/*O*-glycans and HMOs. Our study demonstrates that a difference in *O*-glycan composition and concentration but not in those of *N*-glycan is present in individuals based on their secretor status.

## 2. Results and Discussion

### 2.1. Feasibility Study of Relative Quantitation of N/O-Glycans via Label-Free LC–MS

We examined the feasibility of using relative quantitation via label-free LC–MS methods with bovine fetuin as a model glycoprotein and slightly modified reported methods to analyze *N*-glycans [17,31]. Briefly, detergent-denatured fetuin was digested using PNGase F, and the released *N*-glycans were analyzed via LC–MS and LC–MS/MS without further labeling. Consequently, 28 complex-type *N*-glycans were detected via LC–MS, and their structures were determined via MS/MS analysis and were consistent with those in a previous report [32]. The LC–MS measurements showed a high linearity from 10 to 100 μg/mL fetuin (correlation coefficient, r > 0.995, Appendix A). Thus, the label-free LC–MS method was applicable to the relative quantitation of *N*-glycans.

To simplify the *O*-glycan analysis, we verified whether chemically released *O*-glycans could be detected and relatively quantified as oximated derivatives without the labeling reaction via LC–MS. *O*-glycan was released from fetuin using the releasing reagent containing NH_2_OH and DBU, and the oximated *O*-glycans were analyzed via LC–MS without further labeling. Three signals of *m*/*z* 690.25, 981.34, and 1346.48 were observed in the LC–MS chromatogram, and their structures were analyzed via LC–MS/MS. In the MS/MS spectrum at *m*/*z* 1346.48 (Appendix A), a peak of oximated HexNAc at the reducing end (*m*/*z* 237.09) was observed, resulting from the sequential fragmentation of the glycan corresponding to the elimination of two Neu5Ac (−291 Da), two Hex (−162 Da), and one HexNAc (−203 Da). This MS/MS spectrum indicated a disialylated core2 *O*-glycan (Neu5Acα2,3Galβ1,3(Neu5Acα2,3Galβ1,4GlcNAcβ1,6)GalNAc) with oximation at the reducing end of GalNAc. The MS/MS spectra at *m*/*z* 690.25 and 981.34 also indicated oximated core1 monosialylated and disialylated *O*-glycans, respectively. These *O*-glycans were consistent with reported *O*-glycans of fetuin [29], indicating that the structural analysis of oximated *O*-glycans was feasible. The dependence of the LC–MS area of each oximated *O*-glycan on the fetuin concentration was then confirmed and showed high linearity (10–100 μg/mL fetuin; correlation coefficient, r > 0.988, Appendix A). Thus, the label-free LC–MS analysis of oximated *O*-glycans was applicable for the relative quantitation of *O*-glycans.

### 2.2. Detection of N/O-Glycans in Human Milk via Label-Free LC–MS

*N*-glycans in human milk were enzymatically released and analyzed via label-free LC–MS using the pooled milk, which was mixed in an equal volume of 200 samples of human milk collected in our cohort study [30]. In total, 73 *N*-glycans were detected based on the monosaccharide composition calculated from the observed *m*/*z* (Table 1). Five of these *N*-glycan structures were classified as high-mannose type (N03, N06, N12, N22, and N33), seven as hybrid type (N14, N15, N24–N26, N32, and N34), 59 as complex type (N04, N05, N07–N11, N13, N16–N21, N23, N27–N31, and N35–N73), and two as others (N01 and N02). Based on the different glycosyl modifications, the grouping of these structures included 19 *N*-glycan structures that were both fucosylated and sialylated glycans, 20 structures that were fucosylated glycans only, 15 structures that were sialylated glycans only, and 19 structures that were nonfucosylated and nonsialylated type glycans (Figure 1A).

Similarly, after label-free LC–MS analysis of oximated *O*-glycans in the pooled human milk, 33 *O*-glycans were detected as monosaccharide compositions that matched the observed and calculated *m*/*z* from the glycan composition (Table 2). Based on the different glycosyl modification, the grouping of these structures included three *O*-glycans that were both fucosylated and sialylated glycans, thirteen structures that were fucosylated glycans, twelve structures that were sialylated glycans only, and five structures that were nonfucosylated and nonsialylated glycans (Figure 1B).

*N*/*O*-glycan structures are highly complex and are characterized by various isomers. Here, human milk *N*/*O*-glycans with the same glycan composition but different retention times in HPLC were detected (N10/N11, N18/N19, N20/N21, N25/N26, N40/N41, N45/N46, N54/N55, and N61/N62 in Table 1; O02/O03, O04/O05/O06/O07, O10/O11/O12, O14/O15/O16, and O30/O31 in Table 2), demonstrating the complex and diverse structures of *N*/*O*-glycans in human milk. In total, we detected 73 *N*-glycans containing 34 sialylated (47%) and 39 fucosylated (53%) glycans and 33 *O*-glycans containing 15 sialylated (45%) and 16 fucosylated (48%) glycans. More than 70% of the detected *N*/*O*-glycans were sialylated, fucosylated, or both sialylated and fucosylated (Figure 1). The total numbers of *N*/*O*-glycans detected via label-free LC–MS were comparable with those reported previously [21,23,24,27]. We did not consider sulfated *O*-glycans in this study, although a previous study had reported them [24]. These results are consistent with those reported in previous studies [17,21,33], emphasizing the significantly higher levels of fucosylation and sialylation in both *N*- and *O*-glycans in human milk. However, our results deviate from the lower proportions documented by Wang et al. in healthy individuals [27]. This discrepancy suggests the possibility of variation due to lactation period, ethnicity, and regional factors and the need for a nuanced understanding of human milk glycosylation. This observed variation highlights the complex nature of glycan in human milk and warrants further detailed investigation.

### 2.3. Relative Quantitation of N/O-Glycans in Human Milk via Label-Free LC–MS

The relative quantitation of *N*/*O*-glycans in human milk using the label-free LC–MS methods was evaluated using the dilution assay with sequential diluted pooled milk. The relative MS area of 73 *N*-glycans and 33 *O*-glycans had an excellent linear relationship (r > 0.8) with the dilution rate of the pooled milk samples, demonstrating that the label-free LC–MS could be used for the relative quantitation of most of the *N*/*O*-glycans detected in this study (Table 1 and Table 2).

### 2.4. Structural Analysis of N/O-Glycans in Human Milk via MS/MS Analysis

The MS/MS analysis using SimGlycan software 2.0 (score > 70) was combined with manual inspection to propose reliable structures for 17 *N*-glycans (N03, N05, N06, N07, N12, N17, N22, N28, N31, N33, N36, N37, N42, N51, N57, N58, and N66) and 12 *O*-glycans (O01, O08, O09, O17, O19, O20, O22, O24, O25, O26, O27, and O28) in pooled human milk (Figure 2). The sialic acid (Neu5Ac) and fucose (Fuc) linkage of these *N*/*O*-glycans was not determined. The seventeen *N*-glycans were composed of five high-mannose-type glycans (N03, N06, N12, N22, and N33), eleven biantennary complex-type glycans (N17, N28, N31, N36, N37, N42, N51, N57, N58, and N66), and one monoantennary complex-type glycan (N07). Of the seventeen *N*-glycans, four were fucosylated only (N07, N28, N31, and N37), two were sialylated only (N36 and N51), and four were both fucosylated and sialylated (N42, N57, N58, and N66). No *N*-glycans with Fuc linked to galactose (Gal) were characterized. The twelve *O*-glycans included one core1 *O*-glycan (O01), ten core2 *O*-glycans (O09, O17, O19, O29, O22, O24, O25, O26, O27, and O28), and one core3 *O*-glycan (O08) while four were fucosylated only (O01, O08, O20, and O26), three were sialylated only, (O19, O25, and O27), and two both fucosylated and sialylated (O24 and O28). Structures with Fuc linked to both Gal and *N*-acetylglucosamine (GlcNAc) were characterized. All *N*/*O*-glycans with proposed structures had a high linearity in the dilution assay (r > 0.8, Table 1 and Table 2). However, one high-mannose-type *N*-glycan (N06 in Table 1) was excluded from the quantitative analysis of the subsequent 200 human milk samples because correct quantitation was prevented by the overlap of an unidentified component with the same *m*/*z* and retention time as N06 in the LC–MS analysis. Therefore, to quantify N06, the released *N*-glycans must be purified to remove the unknown components before LC–MS analysis.

### 2.5. Relative Quantitation of N/O-Glycans in 200 Samples of Japanese Human Milk via Label-Free LC–MS

Using the label-free LC–MS method, we quantified 16 *N*-glycans and 12 *O*-glycans with proposed structures except N06 in milk samples from 200 Japanese women. The quantification was performed by assessing the LC–MS peak area for each glycan. The abundance of each *N*/*O*-glycan was corrected using an internal standard (Appendix A). To facilitate the comparison of *N*/*O*-glycan distributions among individuals, the corrected MS areas were normalized, as detailed in Appendix A and displayed in Appendix A. The median value for the concentration of each glycan was lower than the respective mean value, indicating that the distribution of each glycan concentration was nonsymmetric and positively skewed. For most glycans, some samples had a 5–20-fold higher concentration than the mean value. The coefficient of variation (CV) of each glycan concentration was 77–252% for *N*-glycan and 74–175% for *O*-glycan in the 200 individual samples of human milk.

The absolute concentrations of the 11 representative HMOs in the same 200 samples of human milk were measured using HPAE-PAD to compare the concentration distribution of *N*/*O*-glycans with that of HMOs (Appendix A). Each concentration range was consistent with previously reported values [34,35]. The CV of each HMO was 43–114% (Appendix A). Comparing the CVs of *N*/*O*-glycans with those of HMOs, the concentrations of both *N*/*O*-glycans in human milk had a more significant distribution among individuals than those of HMOs.

The concentration of HMOs can vary significantly among Individuals. In this study, the *N*/*O*-glycans varied more significantly in concentration than the HMOs did. Although this large variance may be due to the variance in protein concentration in human milk since *N*/*O*-glycans are conjugated to proteins, we believe that the significant variance in *N*/*O*-glycan concentration is independent of the variance in protein concentration because the CV of the protein concentration in the 200 samples of human milk was 19.8%, which was significantly smaller than that of *N*/*O*-glycans. Factors, such as the amount of sugar nucleotide substrates and the activities of glycosyltransferases and glycosylhydrolases, influence the concentration of *N*/*O*-glycans [36]. Further studies are, therefore, needed to elucidate the cause of the higher variance in *N*/*O*-glycans.

### 2.6. Cluster Analysis of N/O-Glycans and HMOs in 200 Human Milk Samples

Hierarchical clustering was used to visualize the quantitative data of 16 *N*-glycans, 12 *O*-glycans, and 11 HMOs from 200 samples of human milk on a heatmap. These *N*/*O*-glycans and HMOs were classified into three main groups according to the dendrogram (Group A, B, and C; Figure 3). Group A, which comprised all *N*-glycans and four HMOs, was divided into three subgroups (A1, A2, and A3). Subgroup A1 contained six sialylated complex-type *N*-glycans (N36, N42, N51, N57, N58, and N66) and two nonsialylated complex-type *N*-glycans (N17 and N37). Subgroup A2 included three nonsialylated complex-type *N*-glycans (N05, N28, and N31) and four high-mannose-type *N*-glycans (N03, N12, N22, and N33). All sialylated *N*-glycans belonged to subgroup A1 and none to subgroup A2. Subgroup A3 contained nonfucosylated HMOs (3′-SL, 6′-SL, LNT, and LNnT). Group B contained three *O*-glycans (O1, O20, and O28) and four HMOs (2′-FL, LDFT, LNFP-I, and LNDFH-II) where Fuc was linked to Gal, presumably with an α1,2-bond. Group B was further divided into two subgroups according to the presence of one (O1, 2′-FL, or LNFP-I) or two (O20, O28, LDFT, and LNDFH-I) Fuc in the glycan composition. Group C consisted of three *O*-glycans with Fuc linked to GlcNAc (O08, O24, and O26), six *O*-glycans without Fuc (O09, O17, O19, O22, O25, and O27), and three fucosylated HMOs (3′-FL, LNFP-II, and LNDFH-II) with Fuc linked to Glc or GlcNAc.

All *N*-glycans were classified into group A and appeared to be subgrouped depending on the presence or absence of sialic acid in their structures, suggesting that the profile pattern of *N*-glycans in human milk was related to activity of sialyltransferase. *O*-glycans and fucosylated HMOs were classified into group B or C depending on whether Fuc was linked to Gal. The concentration of HMOs classified into group B and C is known to considerably differ between secretor and nonsecretor milk [9], whereas that in group A is less sensitive to secretor status, implying that classification into these three groups is dependent on the maternal secretor status. Therefore, *O*-glycans may have been classified into group B or C according to secretor status. These results suggested that the concentration of some *O*-glycans, as well as HMOs, is associated with secretor status.

### 2.7. Comparison of N/O-Glycans and HMOs between Secretor Status

The relative concentration of *N*/*O*-glycan and HMOs was compared between secretor and nonsecretor milk (Figure 4, Appendix A). All milk samples were categorized as secretor milk (2′-FL > 50 mg/L) or nonsecretor milk (2′-FL < 50 mg/L) according to the 2′-FL concentration [37]. Of the 200 human milk samples, 161 (80.5%) were secretor milk and 39 (19.5%) were nonsecretor milk. Among the *N*-glycans and HMOs classified in group A, the concentration of four *N*-glycans (N12, N28, N42, and N58) and LNnT were significantly higher in secretor milk than that in nonsecretor milk, whereas the concentration of two *N*-glycans (N31, N36) and LNT was significantly lower than that in secretor milk. The concentration of all *O*-glycans and HMOs classified in group B, where Fuc is linked to Gal, was higher in secretor milk than in nonsecretor milk. Among the *O*-glycans and HMOs classified in group C, the concentration of five *O*-glycans (O8, O9, O24, O26, and O27) and three HMOs (3-FL, LNFP-II, and LNDFH-II) was higher in nonsecretor milk than that in secretor milk. None of the *O*-glycans or HMOs classified in group C had higher concentrations in secretor milk.

All the *O*-glycans that were more abundant in secretor milk possessed Fuc linked to Gal and belonged to group B. Although HMOs with Fuc linked to Gal via FUT2 (2′-FL, LDFT, LNFP-I, and LNDFH-II) were rarely found in nonsecretor milk, *O*-glycans with Fuc linked to Gal such as O1, O20, and O28 were present in nonsecretor milk, albeit at low concentrations. Several of these *O*-glycans may be derived from blood glycoprotein. Alternatively, fucosyltransferase 1 (FUT1) could be involved in synthesizing these *O*-glycans in human milk [38,39,40]. Conversely, all *O*-glycans that were more abundant in nonsecretor milk did not possess Fuc linked to Gal and belonged to group C. For *N*-glycans, the secretor status was unrelated to the presence or absence of Fuc in their structure. The findings indicate that secretor status has a notable impact not only on the composition and concentration of HMOs but also on those of *O*-glycans, instead of *N*-glycans.

In our results, the concentration of *O*-glycans and HMOs was significantly different between secretor and nonsecretor. However, although the concentration of some *N*-glycans was different between secretor and nonsecretor, no specific association between the structures of *N*-glycans and secretor status was observed. Contrary to our observations, previous reports have emphasized the impact of secretor status on *N*-glycans [21,22,23], highlighting a higher abundance of fucosylated and sialylated glycans in secretor milk. The discrepancy with previous reports may be because the *N*-glycans analyzed in this study did not include *N*-glycans with Fuc linked to Gal. Furthermore, the concentration of *O*-glycans have been reported to be higher in secretor milk [24], which is not entirely consistent with our results. This discrepancy may be due to differences in the ethnicity of the human milk donor used in this study or difference in sample size compared to previous reports. Secretor status affects infant health and development through changes in glycan composition [41,42]. Differences in glycan structure have different effects on the bifidobacterial growth [19,22,43], infant gut microbiota [44], and immune response [45]. Further studies are needed to elucidate the specific effects of glycan profiles in human milk on infants. This study may provide a basis for future research on the functional implications of such differences in glycan profiles and their potential impact on infants.

## 3. Materials and Methods

### 3.1. Human Milk

Human milk samples were provided by participants in the Japanese Human Milk Study registered in the University Hospital Medical Information Network Clinical Trials Registry (UMIN ID 000015494, 2014) [30]. A total of 1210 lactating women were recruited; 83 did not provide valid responses, and 5 met the exclusion criteria, i.e., the women or their partners were not Japanese. Thus, 1122 women were included in our cohort study, of whom 930 provided sufficient (>50 mL) human milk at 1–2 months postpartum. This study used 200 human milk samples randomly selected from milk samples at 1–2 months postpartum provided by 930 participants. The milk samples were pooled by mixing equal volumes of the selected 200 milk samples and were stored at −80 °C.

### 3.2. Reagents

The EZ-Glyco O-glycan Prep kit was purchased from Sumitomo Bakelite (Tokyo, Japan). PNGase F (including protein denaturing buffer, NP-40, and glycobuffer) was purchased from New England Biolabs (Ipswich, MA, USA). Vivaspin 500 concentrators were purchased from Sartorius (Göttingen, Germany). Water and acetonitrile for LC–MS were purchased from Kanto Kagaku (Tokyo, Japan). Pure-grade HMO standards were purchased from Carbosynth Ltd. (Compton, UK). Other reagents were purchased from standard vendors.

### 3.3. Preparation of N-Glycans for Label-Free LC–MS Analysis

We combined and slightly modified previously reported methods [17,31] to enable label-free LC–MS analyses of *N*-glycans in a large number of human milk samples. Briefly, 3 mol/L sodium borohydride solution (10 μL) was added to a fetuin solution (20 μL) or human milk (20 μL) and left at ambient temperature for 30 min. Acetic acid (10 μL) was added to neutralize the mixture prior to incubation for 30 min at ambient temperature. Subsequently, the protein-denaturing buffer (10 μL) was added and the solution was heated at 100 °C for 10 min. Then, 1 mol/L ammonium bicarbonate solution (10 μL) and 123 mmol/L iodoacetamide (10 μL) were added, and the solution was incubated at ambient temperature for 1 h in the dark. Following the addition of NP-40 (10 μL), the reaction mixture was incubated with PNGase F (3 units) at 37 °C for 16 h. Subsequently, 450 nmol/L maltohexaose solution (125.5 μL), as an internal standard for LC–MS measurement, was added and centrifuged at 15,000× *g* for 10 min using a 10,000 molecular weight cut-off ultrafiltration membrane (Vivaspin 500) to remove proteins. The permeate was diluted twice using acetonitrile for LC–MS measurement.

### 3.4. Preparation of O-Glycans for Label-Free LC–MS Analysis

An EZ-Glyco O-glycan Prep kit was used for *O*-glycan analysis. Fetuin solution (5 μL) or human milk (5 μL) was mixed with the Glycan Release Reagent (Reagent A:B = 2:1, 7.5 μL) and incubated at 50 °C for 20 min [29]. Subsequently, 450 nmol/L maltohexaose solution (107.5 μL) was added and centrifuged at 15,000× *g* for 10 min with Vivaspin 500 to remove proteins. The permeate was diluted twice using acetonitrile for LC–MS measurement.

### 3.5. LC–MS Ananlysis of N/O-Glycans

*N*/*O*-glycans of fetuin or human milk were analyzed using a Q-Exactive mass spectrometer coupled to an UltiMate 3000 (Thermo Fisher Scientific, Waltham, MA, USA). The *N*- or *O*-glycan solution (10 μL) was applied to a Glycanpac AXH-1 column (2.1 mm i.d. × 150 mm; 3 μm particle size; Thermo Fisher Scientific) at 40 °C. The mobile phase comprised acetonitrile (solvent A) and 50 mmol/L ammonium formate (pH 4.4) (solvent B) in a gradient elution of 0 to 55 min at 10–35% and then 55 to 65 min at 35% B. The flow rate was 0.4 mL/min. The electrospray voltage and heat capillary temperature were 3.5 kV and 275 °C, respectively. Nitrogen (99.5% purity) was used as sheath gas (set to 35), auxiliary gas (set to 10), and collision gas. For the relative quantitation of *N*/*O*-glycans, full-scan mass spectra were acquired in positive ion mode from *m*/*z* 400 to 2000 and a resolution of 70,000. For structural analysis of *N*/*O*-glycans, full-scan mass spectra were acquired in automatic data-dependent mode using one precursor scan, followed by five MS/MS scans. The LC–MS/MS spectra were acquired using a higher energy collisional dissociation of *m*/*z* 400–2000 and resolution of 17,500 and stepped normalized collision energies of 10, 20, and 30.

### 3.6. HMO Analysis via HPAE-PAD

HPAE-PAD was used to analyze 11 representative HMOs in human milk: 2′-FL, 3-fucosyllactose (3-FL), 3′-sialyllactose (3′-SL), 6′-sialyllactose (6′-SL), lacto-*N*-tetraose (LNT), lacto-*N*-neotetraose (LNnT), lactodifucotetraose (LDFT), LNFP-I, lacto-*N*-fucopentaose-II (LNFP-II), lacto-*N*-difucohexaose-I (LNDFH-I), and lacto-*N*-difucohexaose-II (LNDFH-II). Human milk (20 μL) was diluted 30-fold with water (580 μL) and centrifuged at 15,000× *g* for 10 min using Vivaspin 500 concentrators to remove proteins. The permeate was used for HPAE-PAD measurement without further dilution. The HMO solutions were applied to a Dionex CarboPac PA1 column (4 mm i.d. × 250 mm; 10 μm particle size; Thermo Fisher Scientific) at 30 °C. The mobile phase comprised water (solvent C), 200 mM NaOH (solvent D), and 600 mM sodium acetate in 100 mM NaOH (solvent E) in the following gradient elution: 0–7 min at 45% D and 10% E; 7–10 min at 45–37.5% D and 10–25% E; 10–20 min at 37.5% D and 25% E; 20–20.1 min at 37.5–45% D and 25–10% E; and 20.1–25 min at 45% D and 10% E. The flow rate was 1 mL/min.

### 3.7. Data Treatment

LC–MS raw data were analyzed using Compound Discoverer 2.1 software (Thermo Fisher Scientific) to detect *N*/*O*-glycans and obtain the LC–MS area of each *N*/*O*-glycan. Compound Discoverer parameters were set to the following: mass range, 0–5000 Da; mass tolerance, 5 ppm; intensity tolerance, 30%; S/N threshold, 3; min. element counts, CH; max. element counts, C200H400N40O200; and adduct ions, [M+H]^+^, [M+Na]^+^, [M+K]^+^, [M+2H]^2+^, [M+H+Na]^2+^, [M+H+K]^2+^, [M+2Na]^2+^, [M+Na+K]^2+^, and [M+2K]^2+^. In-house mass lists were used based on previous reports of human milk glycans [15,17,18,19,20,21,24,25,26,27]. The Fill Gaps node was used to obtain the LC–MS area of glycans in all samples. The LC–MS area of each *N*/*O*-glycan was corrected using that of maltohexaose as an internal standard. LC–MS/MS spectra were computationally analyzed to characterize *N*/*O*-glycan structures using SimGlycan 2.0 software (PREMIER Biosoft, Palo Alto, CA, USA). The score criterion for structure proposal was set at >70. The structural assignment was manually validated. HPAE-PAD data were analyzed using Chromeleon software ver. 6.8 (Thermo Fisher Scientific). The concentrations of 11 HMOs were absolutely determined using HMO standards. All milk samples were categorized as secretor milk (2′-FL > 50 mg/L) or nonsecretor milk (2′-FL < 50 mg/L) according to the 2′-FL concentration [37].

### 3.8. Statistics

For the relative quantitation of *N*/*O*-glycans in 200 human milk samples, the corrected LC–MS areas for each *N*/*O*-glycan were normalized to a mean of 1. For hierarchical cluster analysis, the corrected LC–MS areas were standardized to a mean of 0 and a standard deviation of 1. All statistical analyses were performed in R version 3.6.2 and RStudio program (version 2023.03.0, Integrated Development for R. RStudio, PBC, Boston, MA, USA).

### 3.9. Ethics

The study was conducted according to the recommendations of the Ethical Guidelines for Clinical Research (Ministry of Health, Labor, and Welfare, Japan) and was approved by the Ethics Committee of the Fukuda Internal Medicine Clinic (IRB No. 20140621-03). Written informed consent was obtained from participants in accordance with the Declaration of Helsinki.

## 4. Conclusions

Herein, we used label-free LC–MS methods to relatively quantify *N*/*O*-glycans in 200 samples of human milk at 1–2 months postpartum. This quantitative analysis of *N*/*O*-glycans in human milk demonstrated that secretor status not only influences the composition and concentration of HMOs but also those of *O*-glycans rather than those of *N*-glycans in human milk. We analyzed 200 individual human milk samples, whereas previous reports analyzed pooled milk samples or <10 individual milk samples. To our knowledge, this is the first comprehensive study of *N*/*O*-glycans in a large number (200) of human milk samples and provides important insights into the understanding of *N*/*O*-glycans in human milk. The relationship between the concentration of *N*/*O*-glycans in human milk and infant health and growth is particularly interesting. These questions will be addressed in future studies as information on infant health and development was also collected in this human milk cohort study.

## Figures and Tables

**Figure 1 ijms-25-01772-f001:**
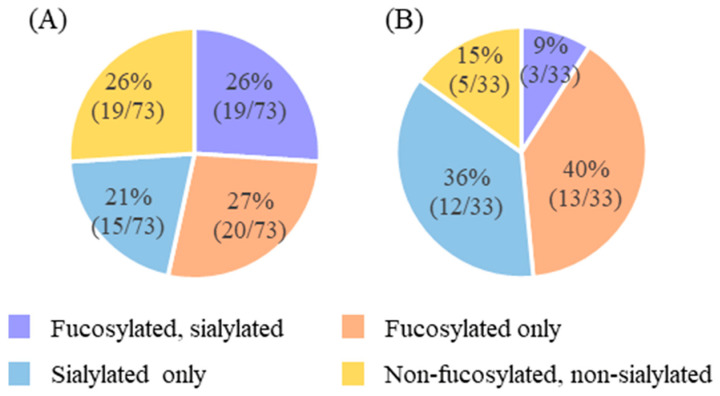
Modification features of *N*-glycans (**A**) and *O*-glycans (**B**) detected in human milk via label-free LC–MS.

**Figure 2 ijms-25-01772-f002:**
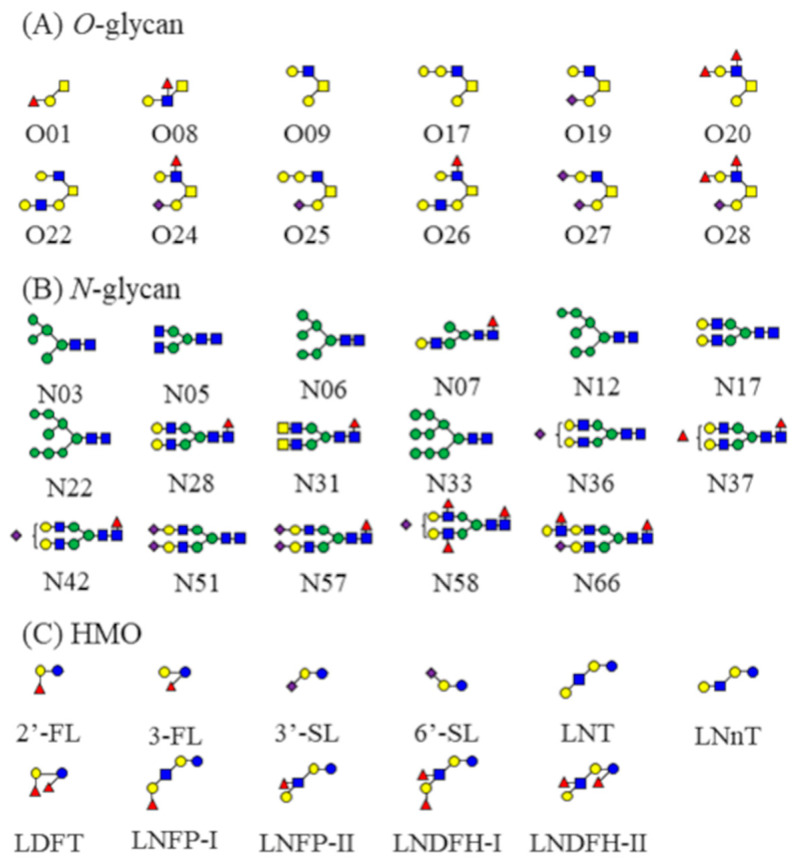
Structures of *N*/*O*-glycans and HMOs. The structures of 17 *N*-glycans and 12 *O*-glycans were characterized via LC–MS/MS analysis of composite human milk samples. The letters under each structure correspond to numbers in Table 1 and Table 2. Color symbols are as follows: green circle, mannose; yellow square, GalNAc; yellow circle, Gal; blue square, GlcNAc; purple diamond, Neu5Ac; and red triangle, Fuc.

**Figure 3 ijms-25-01772-f003:**
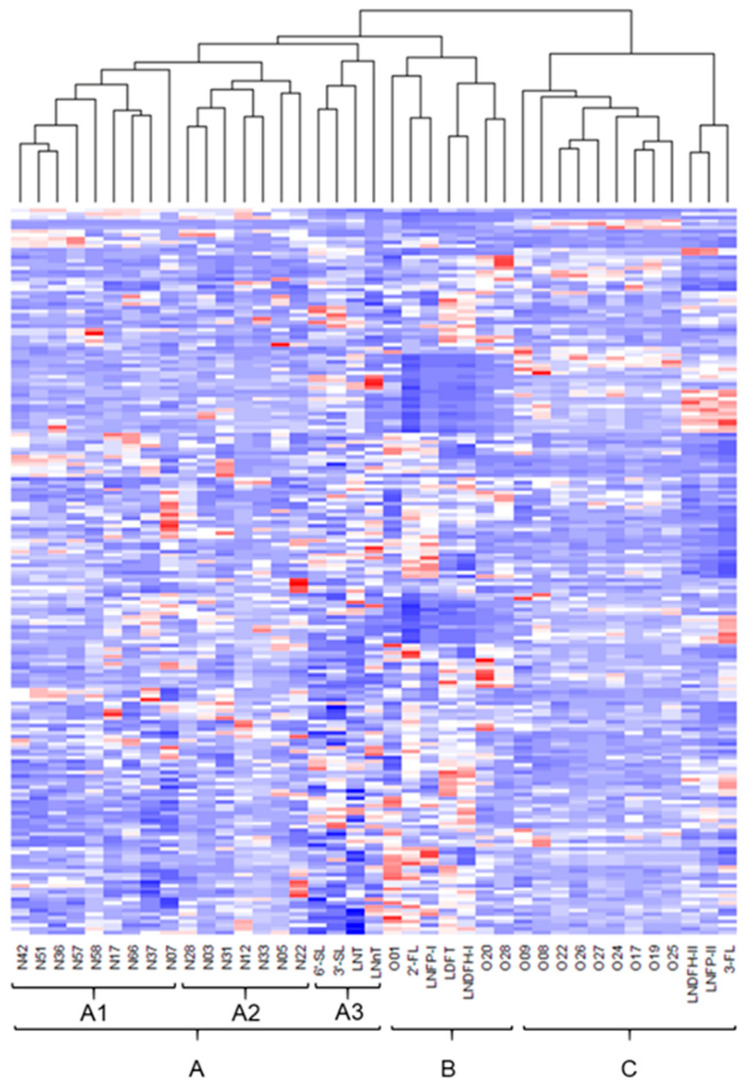
Heatmap of the relative concentration of *N*- and *O*-glycans and HMOs (columns) in 200 samples of human milk (rows) with dendrogram. The relative concentration of each glycan was standardized to a mean of 0 and a standard deviation of 1. Colors range from blue (minimum relative concentration) to red (maximum relative concentration) and white represents the mean value. The letters below the heatmap correspond to each glycan in Figure 2. The lower letters represent the main groups (A, B, C) and subgroups (A1, A2, A3) classified by the dendrogram.

**Figure 4 ijms-25-01772-f004:**
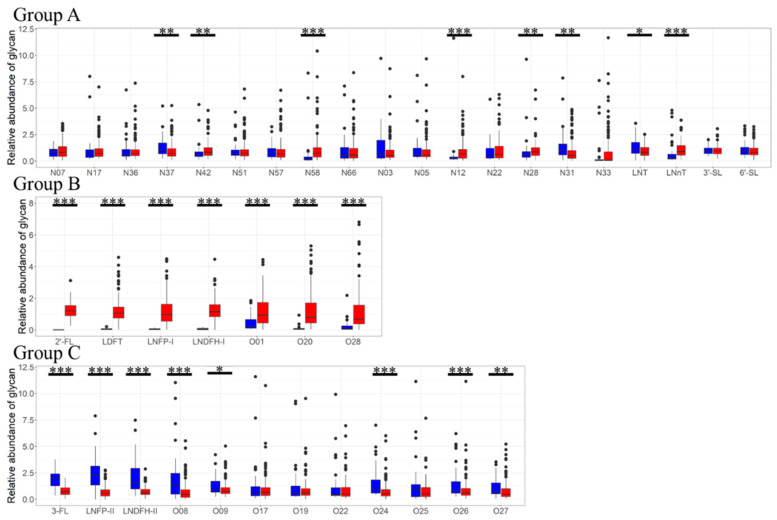
Comparison of the relative concentration of *N*/*O*-glycans and HMOs between secretor milk (red, *n* = 161) and nonsecretor milk (blue, *n* = 39). The relative concentration of each glycan was normalized to a mean of 1. * *p* < 0.05, ** *p* < 0.01, and *** *p* < 0.001 (Mann–Whitney U test).

**Table 1 ijms-25-01772-t001:** *N*-glycans detected in human milk.

No.	RT /min	Mass	Monosaccharide Composition				r *	No.	RT /min	Mass	Monosaccharide Composition				r *
			**Hex**	**HexNAc**	**Fuc**	**Neu5Ac**					Hex	HexNAc	Fuc	Neu5Ac	
N01	25.4	1056.38	3	2	1		0.987	N38	47.1	1948.70	6	4	1		0.989
N02	32.3	1218.44	4	2	1		0.969	N39	52.1	1964.71	7	4			0.985
N03	32.4	1234.43	5	2			0.917	N40	42.9	1972.70	4	5		1	0.970
N04	34.1	1275.43	4	3			0.903	N41	42.3	1972.70	4	5		1	0.974
N05	35.3	1316.46	3	4			0.977	N42	45.2	2077.75	5	4	1	1	0.933
N06	35.9	1396.49	6	2			0.979	N43	42.4	2094.76	6	4	2		0.971
N07	38.5	1421.52	4	3	1		0.959	N44	48.3	2110.79	7	4	1		0.992
N08	34.6	1462.54	3	4	1		0.932	N45	47.3	2118.75	4	5	1	1	0.979
N09	34.0	1478.54	4	4			0.893	N46	44.5	2118.76	4	5	1	1	0.977
N10	38.8	1519.54	3	5			0.956	N47	44.4	2134.73	5	5		1	0.987
N11	35.8	1519.55	3	5			0.956	N48	44.4	2151.77	6	5	1		0.966
N12	38.0	1558.54	7	2			0.803	N49	46.2	2192.80	5	6	1		0.970
N13	40.6	1566.56	4	3		1	1.000	N50	46.3	2208.78	6	6			0.996
N14	38.1	1583.58	5	3	1		0.994	N51	51.1	2222.79	5	4		2	0.964
N15	38.1	1599.57	6	3			0.997	N52	47.0	2223.81	5	4	2	1	0.970
N16	36.8	1624.60	4	4	1		0.902	N53	48.7	2263.80	4	5		2	0.977
N17	37.5	1640.60	5	4			0.947	N54	43.7	2280.79	5	5	1	1	0.902
N18	42.1	1681.59	4	5			0.994	N55	45.9	2280.79	5	5	1	1	0.986
N19	39.2	1681.61	4	5			0.952	N56	46.1	2297.83	6	5	2		0.975
N20	45.1	1712.61	4	3	1	1	0.966	N57	51.8	2368.85	5	4	1	2	0.983
N21	41.5	1712.62	4	3	1	1	0.992	N58	48.1	2369.87	5	4	3	1	0.991
N22	40.3	1720.60	8	2			0.804	N59	53.6	2425.83	5	5		2	0.985
N23	38.1	1722.64	3	6			0.988	N60	48.4	2443.88	6	5	3		0.948
N24	42.6	1728.60	5	3		1	0.942	N61	52.9	2483.89	5	6	1	1	0.988
N25	40.9	1729.61	5	3	2		0.975	N62	47.5	2483.89	5	6	1	1	0.989
N26	38.1	1729.62	5	3	2		0.935	N63	47.5	2499.86	6	6		1	0.995
N27	40.7	1753.63	3	4	1	1	0.991	N64	51.7	2530.89	6	4	1	2	0.982
N28	38.5	1786.65	5	4	1		0.976	N65	52.9	2546.91	7	4		2	0.907
N29	45.8	1802.64	6	4			0.972	N66	50.0	2588.94	6	5	2	1	0.952
N30	40.7	1827.66	4	5	1		0.960	N67	50.1	2589.94	6	5	4		0.983
N31	40.7	1868.70	3	6	1		0.959	N68	54.0	2733.97	6	5	1	2	0.994
N32	43.5	1874.67	5	3	1	1	0.976	N69	51.0	2736.00	8	7			0.997
N33	42.3	1882.63	9	2			0.989	N70	52.7	2790.95	6	6		2	0.984
N34	44.7	1890.66	6	3		1	0.895	N71	51.3	2808.00	7	6	1	1	0.810
N35	42.4	1915.65	4	4	1	1	0.974	N72	59.0	2879.01	6	5		3	0.977
N36	44.3	1931.69	5	4		1	0.965	N73	58.3	3082.04	6	6		3	0.907
N37	44.4	1932.69	5	4	2		0.951								

* Correlation coefficient (r) between the relative LC–MS area of each glycan and the dilution rate of the pooled milk samples in the dilution assay.

**Table 2 ijms-25-01772-t002:** *O*-glycans detected in human milk.

No.	RT /min	Mass	Monosaccharide Composition				r *	No.	RT /min	Mass	Monosaccharide Composition				r *
			Hex	HexNAc	Fuc	Neu5Ac					Hex	HexNAc	Fuc	Neu5Ac	
O01	12.4	544.21	1	1	1		0.986	O18	33.0	980.34	1	1		2	0.998
O02	21.2	689.25	1	1		1	0.912	O19	29.4	1054.38	2	2		1	0.987
O03	18.0	689.25	1	1		1	0.974	O20	25.6	1055.40	2	2	2		0.999
O04	24.0	706.26	2	1	1		0.996	O21	28.5	1071.40	3	2	1		0.994
O05	27.5	706.26	2	1	1		0.994	O22	29.3	1128.42	3	3			1.000
O06	18.8	706.27	2	1	1		0.987	O23	30.1	1183.42	1	2		2	0.999
O07	17.0	706.27	2	1	1		0.965	O24	32.7	1200.44	2	2	1	1	0.998
O08	26.3	747.29	1	2	1		0.981	O25	30.6	1216.43	3	2		1	0.863
O09	20.0	763.29	2	2			0.996	O26	32.8	1274.48	3	3	1		0.994
O10	26.7	851.30	2	1		1	0.994	O27	40.5	1345.48	2	2		2	0.964
O11	20.6	851.30	2	1		1	0.949	O28	34.8	1346.49	2	2	2	1	1.000
O12	21.7	851.30	2	1		1	0.857	O29	35.7	1420.54	3	3	2		0.991
O13	24.6	892.33	1	2		1	0.996	O30	27.5	1493.54	4	4			0.953
O14	21.6	909.34	2	2	1		0.999	O31	35.7	1493.55	4	4			0.994
O15	24.3	909.34	2	2	1		0.999	O32	37.8	1548.55	2	3		2	0.992
O16	25.8	909.34	2	2	1		0.987	O33	39.2	1565.57	3	3	1	1	0.986
O17	20.0	925.33	3	2			0.998								

* Correlation coefficient (r) between the relative LC–MS area of each glycan and the dilution rate of the pooled milk samples in the dilution assay.

## Data Availability

The datasets used and/or analyzed in the current study are available from the corresponding author upon request.

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
