# Peer review of "Label-Free Liquid Chromatography–Mass Spectrometry Quantitation of Relative N- and O-Glycan Concentrations in Human Milk in Japan"

_ijms, 2024, doi:10.3390/ijms25031772_

Round 1
Reviewer 1 Report
Comments and Suggestions for Authors
IJMS 22116
Label-free liquid chromatography–mass spectrometry quantitation of relative N- and O-glycan concentrations in human milk 3 in Japan by Yamaguchi et al.
The research on milk oligosaccharides is authentic. The analytical methods are constantly improved. Quantification of the oligosaccharides stays a challenge, specifically because, up to now, they had to be labelled, which is challenging at the low amounts present. The authors devised a method by which the labelling step was omitted. The authors went on to identify a number of oligosaccharides and quantify them by mass spectrometry. The methods used are state of the art. An important contribution was made to the field. The document is well written with only a few improvements suggested (see below). The document may be accepted for publication.
Please correct the following:
Line 124: …to a fetuin solution…
126: …(10ul) followed by the protein-denaturing buffer (10ul) and the solution…
127: …Glycan Release Agent…
380: …secretor…

See report for a few minor errors
Reviewer 2 Report
Comments and Suggestions for Authors
The manuscript was properly evaluated the human milk oligosaccharides from 200 randomized sample from Japan. The work was essentially accomplished by label-free chromatography–mass spectrometry (LC–MS) technique which improved the validity of the data. It was organized well, however, two things should be considered prior to publication:
1- Please perform a table of ANOVA for the oligosaccharides obtained from the samples,
2- Please provide a valid conflict of interest which the work performed on the human samples.
Comments on the Quality of English LanguageNA
Reviewer 3 Report
Comments and Suggestions for Authors
The article is devoted to a new analytical method for separating and identifying two different types of glycans found in human breast milk. The so-called HMOs have been known for decades for their unique oligosaccharide composition, which is of great importance with its prebiotic action. Still, for glycoproteins, scientific research is quite poor. The achievements of the research are significant, and one of the important aspects is the huge number of samples analyzed, which enables a good statistical analysis.
However, the manuscript needs extensive editing before it can be accepted for publication. Its biggest flaw is its lack of discussion.
1. Introduction: Define more clearly the purpose of the work.
2. Discussion: Since it is not clear from the experiments in the work why different amounts of N-glycans with different glycosyl modifications are found in some cases and O-glycans in others, the authors should find more papers to discuss well after the results.
3. Since the article demonstrates that a difference in O-glycan composition and concentration but not in those of N-glycan is present in individuals based on their secretor status, the reasons for this should be sought in the literature. No comparison with other sources was done. The authors have to expand the discussion by seeking the meaning of the research.
4. The number of references should be significantly increased, citing sources from recent years.
Comments on the Quality of English LanguageMinor editing of English language required.
Round 2
Reviewer 3 Report
Comments and Suggestions for Authors
All required corrections have been made and the paper is suitable for publication in its current form.